# Positive Liquidity Spillovers from Sovereign Bond-Backed Securities

Peter G. Dunne 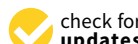

Central Bank of Ireland, Dublin 1 D01 F7X3, Ireland; peter.dunne@centralbank.ie

**Abstract:** This paper contributes to the debate concerning the benefits and disadvantages of introducing a European Sovereign Bond-Backed Securitisation (SBBS) to address the need for a common safe asset that would break destabilising bank-sovereign linkages. The analysis focuses on assessing the effectiveness of hedges incurred while making markets in individual euro area sovereign bonds by taking offsetting positions in one or more of the SBBS tranches. Tranche yields are estimated using a simulation approach. This involves the generation of sovereign defaults and allocation of the combined credit risk premium of all the sovereigns, at the end of each day, to the SBBS tranches according to the seniority of claims under the proposed securitisation. Optimal hedging with SBBS is found to reduce risk exposures substantially in normal market conditions. In volatile conditions, hedging is not very effective but leaves dealers exposed to mostly idiosyncratic risks. These remaining risks largely disappear if dealers are diversified in providing liquidity across country-specific secondary markets and SBBS tranches. Hedging each of the long positions in a portfolio of individual sovereigns results in a risk exposure as low as that borne by holding the safest individual sovereign bond (the Bund).

**Keywords:** safe assets; securitisation; dealer behaviour; liquidity; bid–ask spread

## 1. Introduction

Considerable ambiguity exists regarding the likely liquidity effects of introducing Sovereign Bond-Backed Securities, as proposed by Brunnermeier et al. (2017). The competing forces are (i) a reduced supply of the underlying asset and (ii) improved prospects for the more efficient management of inventory risks. This paper argues that trading costs in country-specific European sovereign bond markets would be constrained by an arbitrage relation as a result of the existence of liquid Sovereign Bond-Backed Securities (SBBS) and this would dominate any negative supply effects.[1] This outcome relies on sufficient diversification and hedging by dealers in a truly European-wide context. In this sense, and in some other respects to be discussed below, the outcome envisaged is a deeply integrated market in which SBBS become the primary focus for price discovery while trading costs in most national markets shrink to within tight bounds of those in SBBS markets.

Brunnermeier et al. (2016) and Brunnermeier et al. (2017) propose the issuance of sovereign bond-backed securities (SBBS) in the euro area with tranches that would be sequentially exposed to losses arising from defaults on any of the underlying individual sovereign securities. The senior tranche would make up the majority of the securitisation and be extremely low risk. The junior tranches would be more sensitive to early signs of defaults, but, at the same time, maintain some of the risk reducing benefits of diversification. In theory, this proposal has the potential to induce bond holders to diversify

---

[1] A wider range of potential effects are considered in the report by the ESRB High-Level Task Force on Safe Assets (2018).

beyond their national sovereign markets and this could substantially weaken the sovereign-bank doom-loop that was prevalent during the Sovereign Debt Crisis in Europe. The question remains whether it would work in practice.

Traditionally, banks hold sovereign bonds as collateral for use in short-term treasury management and in official monetary policy operations. This is encouraged by the fact that such assets are zero risk weighted for capital adequacy purposes. During the Great Recession and euro area Sovereign Debt Crisis, many European banks experienced widespread defaults on their loan portfolios and they required recapitalisation from their governments. Sovereigns simultaneously experienced severe current account deficits and expected imminent high bank bail-out costs which drove their yields to record levels. The higher risk of sovereign default fed back into credit and counter-party risks for the linked banks and this exacerbated the expected bail-out costs for sovereigns. These linked risk exposures generated what was referred to as a "vicious circle" at a euro area summit of heads of governments in 2012. It was also referred to as the "doom loop" by Farhi and Tirole (2018) or as the "diabolic loop" by Brunnermeier et al. (2016).

Proposals to break the bank-sovereign vicious circle have been debated as part of a two-year reassessment by the Basel Committee on Banking Supervision of the Regulatory Treatment of Sovereign Exposures (RTSE) (see Basel Committee on Banking Supervision 2017). The Committee decided to retain zero risk weights on domestic sovereign exposures acknowledging that there was no consensus among supervisors, experts, and economists on alternative policy options to amend the framework. Alternative approaches to breaking the doom loop have therefore appeared and the Sovereign Bond Backed Securitisation proposal by Brunnermeier et al. (2016) is just one of these. Several related proposals are assessed by Leandro and Zettelmeyer (2018) and the analysis below would have some relevance for the 'Blue bond' proposal discussed there since that proposal gives rise to a similarly liquid low risk bond that could be used in hedging by dealers.

An apparent drawback of the SBBS proposal is that it would leave already small and illiquid sovereign bond markets with reduced free float and this would do further harm to liquidity and raise issuance costs. If orders arrive infrequently in smaller sovereign bond markets—and if orders of a type needed to reduce inventories are inelastic with respect to dealer pricing—then inventory positions will be held for longer and providing continuous liquidity would be costly. It is frequently argued that such costs would rise if the national free float is reduced by absorption into a securitisation.

The analysis below suggests that such a conclusion arises only when national markets are considered in isolation. A wider perspective introduces the potential for liquidity spillovers due to hedging and diversification in market making. Rather than reducing liquidity, the effects of SBBS on sovereign bond markets should have more in common with how opportunities to trade 'To-Be-Announced' US Agency Mortgage Backed Securities contracts (TBA-MBS) affect trading costs in Specific-Pools of Mortgage Backed Securities, even those that are not cheapest to deliver.[2]

If SBBS are very liquid, they present dealers with instantaneous access to offsetting positions following trades in any given sovereign market and this would established a valid arbitrage relation to constrain trading costs in those markets. SBBS markets (particularly for the senior tranche) are likely to be deeper and more liquid than smaller euro area sovereign bond markets due mainly to the greater amounts of SBBS in issue relative to what is typical for any individual sovereign. However, this is also thought likely because SBBS markets (having factor-like properties) would acquire benchmark status. Yuan (2005) shows that the presence of benchmarks gives rise to liquidity in related markets.[3]

---

[2] Gao et al. (2017) describe how, in that market, dealers typically hedge inventory risk in their Specific-Pool exposures with offsetting TBA trades and they show that impediments to hedging can reduce such liquidity. More interestingly, they conclude that the presence of TBA markets has very widespread beneficial effects on liquidity significantly beyond the mortgage pools that are cheapest to deliver. This is also traced to the ability to hedge inventory holding risk.

[3] In a related paper, the acquisition of benchmark status in pre-crisis European sovereign bond markets is examined in Dunne et al. (2007). Benchmarks tend to become liquid as they are the location for discovery of the systematic component

An important difference between the SBBS proposal and the situation discussed in Yuan (2005) is that increased supply of SBBS directly implies a reduction in the supply of individual sovereign bonds outstanding. Yuan assumes, in contrast, that benchmarks enhance the likelihood of issuance of corporate bonds and more issuance improves liquidity. However, a positive externality that Yuan's analysis does not take into account is the hedging benefits that dealers can avail of in the presence of liquid benchmarks. It also leaves out post-hedging diversification benefits. These externalities are worthy of further examination.

Although we lack actual evidence of the characteristics of SBBS, it is possible to assess their likely behaviour (or how they would have behaved in the past) through a simulation approach. The present analysis derives SBBS yield estimates using the simulation approach of Schönbucher (2003). This permits an assessment of hedge effectiveness and it shows that dealer inventory positions in most national markets can be, to some extent, hedged by using one or more of the tranches of the sovereign bond-backed securitisation. More importantly, it turns out that what cannot be hedged is largely idiosyncratic and diversifiable by dealers who are active in many markets. Since inventory risk only requires compensation for its systematic component, the liquidity benefits of SBBS are enhanced by diversification.

Hedge effectiveness is assessed by measuring the risk of the optimally-hedged position relative to the unhedged portfolio, as in Bessler et al. (2016). The additional diversification benefits are assessed by assuming that dealers typically have positions in all sovereigns (with both equal weights on individual sovereigns and market-size based weights). A core result from the analysis is that trading costs in SBBS markets would determine limits on the size of trading costs in national sovereign bonds.[4]

The following section outlines the type of hedging behaviour by dealers that is deemed likely when sovereign bond-backed securities exist. The relevant literature is then surveyed. Methodologies used to estimate SBBS yields, to measure time-varying hedge ratios and to assess diversification benefits, are then explained. The discussion of results from the application of these methods follows. The conclusion gives some indication of overall liquidity benefits that would flow from the existence of SBBS based on the recent history.

## 2. Hedging, Arbitrage and Diversification

The benefit of hedging with a highly-liquid, contemporaneously correlated, asset is clear-cut in the extreme case of perfect correlation. Let ask and bid prices in the bond market be denoted (**a**) and (**b**) and those in the SBBS market be (**A**) and (**B**), respectively. Assuming no frictions (i.e., no basis, coordination, execution or timing risks and no variability in market making risks or risk aversion), then arbitrage and competition between dealers should keep the two bid–ask spreads close to each other.[5] Perfect correlation in the underlying values of the two securities and the assumption of instantaneous availability of trading opportunities in the highly liquid asset allow us to subtract the common underlying value changes, **V(t)**, from all bid and ask prices in each period **t**, leaving **a\*, A\*, b\*** and **B\*** as timeless (where starred variables are deviations from the relevant common **V(t)**).

A dealer who acquires a long position at a price (**b**) can immediately sell an equal amount in the SBBS market at price (**B**). This leaves the position hedged against movements in **V** until the bond is sold again at a price (**a**) and the SBBS is simultaneously bought at (**A**). Regardless of common movements in **V**, there is a profit for the dealer of **B\* − b\* + a\* − A\***. This profit is trivially increasing in the difference between the spreads **s-S**. In a competitive market, we would expect such differences

---

of the risk premium (in this case, it is envisaged that the different tranches of SBBS would be benchmarks for credit risks within different categories of the market).

[4] Whether these bounds are sufficient to improve on current trading costs is moot. Even if the costs of hedging with SBBS were to exceed current trading costs in national markets, their use in this way would still be relevant in minimising the extent of any deterioration in trading costs due directly to reductions in the free float as a result of the securitisation.

[5] This also relies, for simplicity, on the assumption that there is symmetry in the positioning of spreads relative to the underlying value. If not, then the proposition that follows applies on average across many trades.

in spreads to be competed away (excluding any extra costs associated with operating in the more general environment). The spread in the bond market will, in this case, be primarily determined by the required bid–ask spread in the SBBS market.

If there is only a partial correlation between the value of the SBBS and that of the bond then the remaining risk must be managed somehow. In this case, a diversification strategy should be very effective since remaining risks would largely be idiosyncratic. The finding of a substantial reduction in risk due to diversification can be regarded as indirect evidence that SBBS behave like systematic risk factors.

Since only the non-diversifiable component of the unhedged risk will require compensation, the link between the size of the spread in the hedge instrument and that in the asset being hedged is relatively unaffected relative to the case of a perfectly correlated hedge. Standard diversification arithmetic implies that, increasing the extent of diversification from one sovereign market to 11 uncorrelated markets with similar-sized risks and activity, would produce more than a 70% reduction in risk exposure. Hence, if dealers are active in providing liquidity across all European sovereign markets that contribute to the securitisation, then any unhedged risks will largely average-out in their trading portfolios.[6]

It is important to draw a distinction between hedging with offsetting positions in an asset possessing highly correlated contemporaneous price movements and the alternative of hedging with a futures contract on the underlying. Futures contracts involve risk premiums that impose direct cost on dealers. There is also a considerable difficulty in correctly matching the expected duration of inventory holding with the expiry time of the associated futures contract.[7] Matching the size of an inventory position with standardized sizes in the futures market may also be an issue. Hedging with CDS contracts is similar to the use of a correlated asset but this analogy is also incomplete. The CDS premium will only be highly correlated with the credit risk of the underlying security. The disadvantage of the CDS when compared with a factor-like hedge is that it does not hedge systematic interest rate risk. CDS trading also tends to be less liquid than the underlying.

## 3. Microstructure Literature

Standard models of market making behaviour assume that bid and offer quotes involve a fixed component to cover order processing costs, an inventory management component (e.g., Amihud and Mendelson 1980; Demsetz 1968; Tinic and West 1972; Ho and Stoll 1981; Ho and Stoll 1983 and Stoll 1978) and an adverse selection component to protect the dealer from losses to more informed traders (e.g., Copeland and Galai 1983; Easley and O'Hara 1987 and Glosten and Milgrom 1985). In the case of sovereign bonds, however, the adverse selection component is regarded as small since nearly all information about underlying value of such bonds is public.[8]

In respect of the inventory imbalance component, Stoll (1978) was one of the first to empirically assess the differential effects of systematic and idiosyncratic risk on dealer pricing. Huang and Stoll (1997) simply assert that the inventory component is mainly related to dealers' portfolio-wide inventory imbalances rather than stock-specific imbalances. They use this fact to separate the inventory component of the spread from the adverse selection component. More recently, Hagströmer et al.

---

[6]    It may also be supposed that this benign outcome would be compromised if the SBBS has a difficult-to-forecast correlation with the bond (i.e., if out of sample hedge ratios turn out to be less efficient than they could have been). This is really a type of operational risk and (assuming forecasts are as efficient as possible ex ante) this also gives rise to mostly idiosyncratic and diversifiable risks.

[7]    Bessler et al. (2016) point out that several futures markets for individual sovereign bonds existed pre-EMU and that the alignment of yields during the years of the Great Moderation was largely responsible for the disappearance of all but the German Bund futures. The Great Financial Crisis and the Sovereign Debt Crisis in Europe ultimately led to the reintroduction of futures on Italian BTPs and French OATs. Futures on Spanish Bonos only reappeared in 2015. Naik and Yadav (2003) examine the use of futures to hedge interest rate risk (undesirable duration) in sovereign bond portfolios of dealers and they find support for the propositions about hedging behaviour by Froot and Stein (1998).

[8]    For example, Naik and Yadav (2003) strongly reject the notion that dealers benefit from their information about orderflow even in the relatively concentrated UK Gilts market.

(2016) develops a structural model for price formation and liquidity supply explaining how inventory pressure and price discovery is influenced by joint trading of the same or a similar asset in a parallel market. This shows that where hedging opportunities exist trading costs are partially determined by characteristics of the hedge instrument rather than the specific asset.

Naik and Yadav (2003) provide extensive evidence of the use of futures by dealers in the UK Gilts market to hedge duration (they do not discuss credit risk hedging). They measure risk exposure in two ways; using value-weighted average duration of the whole portfolio of spot and futures bond positions of 15 dealers and alternatively, a value-weighted average of the duration-weighted hedge ratios (where hedge ratios are betas from regressions of returns on individual bonds on the long-gilt futures return). The two measures are very highly correlated (0.92). They show that dealers maintain a short position on average, but the spot and futures positions diverge in opposite directions from their means. They conclude that dealers actively use the futures contract to hedge their spot exposures in order to maintain a target level of interest rate exposure.

This hedging behaviour is consistent with theoretical predictions of Amihud and Mendelson (1980). The evidence of partial hedging supports the findings of Stulz (1996) and suggests that what is not hedged is idiosyncratic (only interest rate risk is being hedged in this case as the sample did not involve significant fluctuations in UK sovereign credit risk). The Naik and Yadav study goes on to show that hedging varies with the costs of hedging, volatility of holding returns, risk aversion and capital shortage. They also show that bigger dealing banks hedge less but do not seem to make profits from this—implying that their knowledge of a significant fraction of orderflow does not translate into an economic advantage.

## 4. Methodology

### 4.1. Derivation of SBBS Yields

Since sovereign bond-backed securities did not exist in the past we rely on estimates of yields on such instruments based on a simulation approach proposed by Schönbucher (2003)—this approach has already been implemented for the case of the SBBS proposal by De Sola Perea et al. (2017) and the ESRB High-Level Task Force on Safe Assets (2018). It is important to state that a motivating principle for the Schönbucher (2003) method is to retain the properties of the underlying relationship between yields and what they imply for default probabilities—including changing correlations and dynamic dependencies. Hence, the estimated SBBS yields in this case are not just some linear combination of the underlying securities (those used as backing for the securitisation). If that were the case, the linear relation would exactly determine the correlations that we rely on for hedge selection. The Schönbucher approach retains the variable probabilities of default in the underlying securities as well as their time-varying interdependencies.

The simulations that produce the SBBS yields are conducted on a daily basis over the period from soon after the introduction of the euro to the end of 2017. The simulations involve draws from 11 of the main sovereign bond yield-spreads (according to the report of the ESRB Task Force on Safe Assets, this group of sovereigns covers approximately 97.5% of outstanding sovereign debt in the euro area). The time period covered contains a lot of variability in circumstances including; the pre-2008 Great Moderation period, the period of the financial crisis in the wake of the Lehman Brothers default, the euro area sovereign debt crisis of 2009–2012 and the subsequent gradual improvement in euro area sovereign bond markets—particularly those of peripheral member states. The latter part of the sample also overlaps with implementation of unconventional monetary policy in the form of largescale bond purchases when these markets became less liquid and harder to hedge. Figure 1 provides a view of the data for the individual sovereign yield changes that was used in the SBBS yield simulation analysis. The negative of yield changes multiplied by 100 is shown. In each case, the conditional volatility and 1% Value-at-Risk are also provided (where the volatility is estimated using a GJR-GARCH(1,1) and the

latter is implied by the conditional volatility under normality). It is important to note that the scale of the *y*-axis is not constant across the panels of this figure.

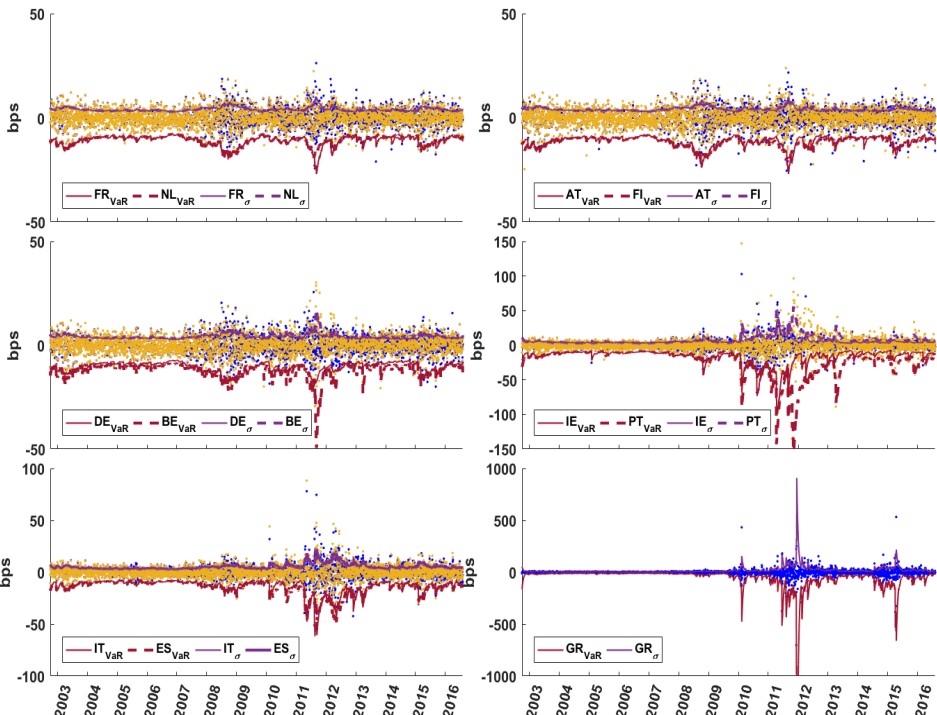

**Figure 1.** Daily Yield-Change Data 11 Sovereigns (bps). This panel of figures displays the daily data of the individual sovereigns that was used in the simulation analysis that generated the Sovereign Bond-Backed Security yields. Each panel contains data for two sovereigns. The negative of the daily yield changes multiplied by 100 are displayed as dots. In each case, the conditional volatility and 1% Value-at-Risk (VaR) are provided (the conditional volatility is estimated using a GJR-GARCH(1,1) and the VaR is the Value-at-Risk implied using the conditional volatility combined with an assumption of normality). It is important to note that the scale of the *y*-axis is not constant across the panels of this figure.

More precisely, the SBBS yield estimation method relies on a simulated default-triggering mechanism interacting with an observable market-based proxy for default probability in the underlying securities (in this case, the default probability proxy is simply the *yield premium* in excess of the lowest yield among the sovereigns). The triggering device generates uniformly-distributed triggers on the unit-interval (where all trigger combinations have cross-correlations of 0.6). Whenever these simulated unit-interval triggers exceed the non-default/survival probability, (1 minus the *yield premium*), losses are calculated as though defaults have occurred.

Each simulation produces simulated default losses among the underlying bonds. These are summed and allocated sequentially to the SBBS securities according to their level of subordination (only spilling over to a more senior tranche if simulated losses have exceeded the total par value of all subordinates). The sum of the yield premiums of the national bonds, for each simulated day, is then allocated to the yield premiums of SBBS according to their proportional allocation of simulated default losses. Hence, the likelihood of triggering a simulated default is determined by the size of

yield premiums and by how correlated the triggers are. However, risk aversion also has some role in determining the premium.[9]

In this way, the probable daily yields on the SBBS components are generated for two different securitisation structures over roughly a 17-year historical period without the need for a structural modelling of the complex dependencies among the underlying sovereigns (e.g., as in Lucas et al. (2017)). This then enables estimation of optimal hedge ratios, hedge effectiveness measures and assessments of the diversification benefits. For reasons of data availability, the simulation is based on yield data for two-, five- and ten-year government bonds of Austria, Belgium, Germany, Spain, Finland, France, Greece, Ireland, Italy, the Netherlands and Portugal, following a weighting scheme based on GDP (averaged over 2006–2015). This basket covers approximately 97.5% of the SBBS volume. As a robustness check, the SBBS yield estimations are re-done using a t-copula instead of the Gaussian copula.

Panels A and B of Figure 2, respectively, depict the time series behaviour of yields on SBBS securities (the derivation of which is discussed above) under two alternative tranching assumptions (70:30 and 70:20:10) while panel C shows yields of a selection of individual sovereigns. The period of the European sovereign debt crisis is highlighted and extends from November 2009, when the Greek government indicated its 2009 deficit projection was being revised upward from 5% to 12.7%, until just after Mario Draghi's speech making a commitment to 'do whatever it takes' to prevent the break-up of the euro in late July 2012. All of the 10-year yield data used in the hedge selection and assessment analysis discussed in the results section has been converted to price and then daily holding period returns assuming a duration of nine years.

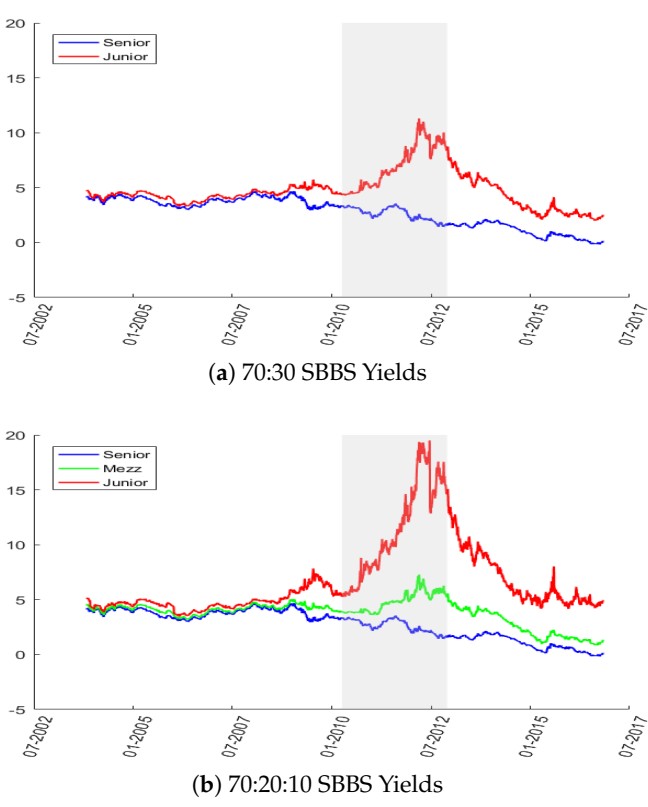

(**a**) 70:30 SBBS Yields

(**b**) 70:20:10 SBBS Yields

**Figure 2.** *Cont.*

---

[9]    The implied risk premium (i.e., yield above the risk-free rate) reflects the risk aversion of the representative market investor on any given day and, hence, may exceed the expected loss anticipated by a risk-neutral investor. This degree of risk aversion enters the simulation and is consequently also reflected consistently in the resulting estimated yields of senior, mezzanine and junior SBBS.

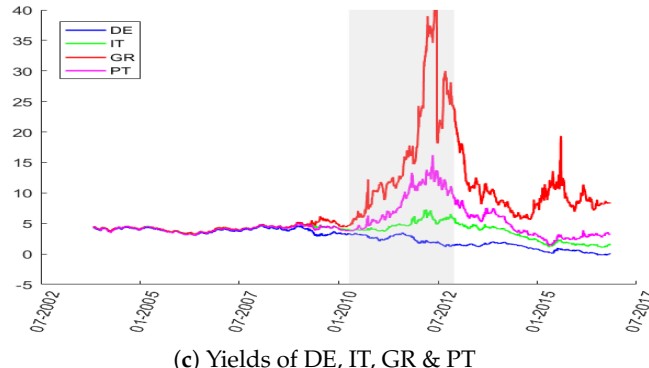

(**c**) Yields of DE, IT, GR & PT

**Figure 2.** Estimated yields on SBBS tranches & selected sovereigns (%). Note: This panel of figures indicates how SBBS yields are likely to have changed over the sample period. These can be compared with a selection of sovereign yields. The shaded area is the euro area Sovereign Debt Crisis period (November 2009–August 2012).

### 4.2. Methodology for Optimal Hedge Selection

There is a well developed literature dealing with the selection of optimal hedge ratios. Chen et al. (2003) and Lien and Tse (2011) provide extensive reviews of different theoretical approaches to determining optimal hedge ratios. A prominent approach to optimising a hedge position is based on minimising the variance of the returns on the hedged portfolio without regard to optimising the expected returns in some way. This is a particularly suitable approach in the case of hedging to facilitate market making in sovereigns since the objective is to minimise risk for the reward of earning the bid–ask spread (less costs) rather than to improve returns from the underlying investment.

In the single hedge case, the optimal hedge ratio (see Ederington 1979 and Baillie and Myers 1991) is simply the negative of the slope coefficient in a regression of the asset return on the hedge instrument return. Composite hedging has sometimes been found to be more effective than relying on a single hedge instrument. This has been claimed in the extant literature for the case of hedging positions in corporate bonds using a combination of the relevant sovereign bond and a futures position in the relevant equity (e.g., Marcus and Ors 1996), using bonds at particular maturities with futures on a variety of other maturities (e.g., Morgan 2008) and hedging mortgage backed securities with Treasuries at 2-, 5- and 10-year tenors (e.g., Koutmos and Pericli 2000). Garbade (1999) also provides an interesting application of a two-asset hedge for a bond. This is similar to the case of hedging with both the Senior and Mezzanine (or Junior) SBBS.

### 4.3. Measuring Out-Of-Sample Hedge Effectiveness

The hedge selection and measurement carried out in this paper follows Bessler et al. (2016) who conduct a similar analysis in a European sovereign bond context. In that analysis, hedge ratios and effectiveness were estimated using rolling OLS, constant conditional correlation (CCC), dynamic conditional correlation (DCC-GARCH) and a Bayesian based mixture of models. In the current analysis, hedge ratios and hedge effectiveness (using single or multiple SBBS as hedges) are estimated for each of the 11 individual sovereign bonds. Hedge effectiveness in each case is measured by the percentage change in risk achieved through hedging. This is done using two different risk metrics for the hedged and unhedged positions. The first metric is the rolling standard deviations of returns. The second metric is based on Values-at-Risk bounds for the hedged and unhedged positions (i.e., the percentage change in the range between the 5% and 95% Value-at-Risks for hedged and unhedged cases).

Hedge effectiveness is therefore measured by the size of the reduction in risk exposure achieved by hedging (essentially, the risk of the hedged position relative to that of the unhedged position). The optimal hedge ratio and the hedge effectiveness measure typically change through time so hedge effectiveness is assessed on a rolling basis across various sub-samples. Engle (2009) applies a pre-crisis

model of time varying covariance to the problem of hedge selection during the Great Financial Crisis and shows that this improves on static approaches that were the industry norm at that time. The rolling linear regression results discussed below are therefore unlikely to overstate the achievable hedge effectiveness.

To keep the exposition tractable, the results obtained using the rolling linear regressions represent a base case (and arguably a lower bound on effectiveness) and are the main focus of the results presented below.[10] The optimal hedge for each of the 11 sovereigns (with 10 years to maturity) was estimated on a rolling basis for the case of the following seven hedge instruments/combinations: {Senior, Mezzanine, Junior, (Senior and Mezzanine), (Senior and Junior), (Mezzanine and Junior), (Senior, Mezzanine and Junior) }. In line with previous literature, the optimal hedge is derived by applying the chosen hedge selection method (e.g., linear regression) over a prior 250 day window and rolling the estimation window forward at regular intervals. The hedge ratio is therefore used in an out-of-sample context for the entire interval between the estimation sub-samples. To keep the exposition tractable, hedge ratio results are presented based on a rolling regression at intervals of 25 days.

## 5. Results

### 5.1. Effectiveness of Hedging without Diversification

The hedge effectiveness results, in terms of the percentage change in risk (usually reduction in risk), are shown for each of three sub-sample periods in Tables 1–3 and the daily returns on the hedged and unhedged positions, for the case of hedging with just the senior and for the case of hedging with a mixture of the senior and mezzanine, are shown in Figures 3–5.[11]

The visual evidence indicates that hedging is generally effective in the pre-Sovereign Debt Crisis period in reducing the volatility of returns (with some isolated exceptions). Hedging is not effective for high-risk sovereigns during the height of the SDC but effectiveness returns to some extent during the recovery. In general, the combined hedge works better than the single hedge in the crises and recovery periods. Figure 3 reveals that hedging is quite consistently effective for three countries (Germany (DE), France (FR) and the Netherlands (NL)). In these cases, the composite hedge seems to eliminate the occasional blips present in the single hedge case. Similarly consistent levels of effectiveness are found for AT and FI (not displayed).

Figure 4 shows the cases of (BE, ES and IT). This clearly reveals how idiosyncratic the experiences of each of the high-risk periphery countries was during the crisis (making them difficult to hedge using SBBS). It is interesting that the composite hedge (senior and mezzanine) works better than the single hedge during the crisis and recovery (apart from one particular day). This tends to improve further with the inclusion of the junior SBBS as a hedge instrument (this more general case is not displayed in the figure but can be seen from the tabulated results discussed below).

Figure 5 shows the more volatile cases of (GR, IE and PT). Here again, there is evidence of hedge ineffectiveness during the crisis with improvement only obvious during the recovery for IE and PT. Again, the composite hedge is better than the single hedge during the recovery for these countries and is particularly good in protecting from the more extreme movements. Although hedging is least effective in reducing risk in the case of the smaller markets, their idiosyncratic risk is more amenable to control using diversification strategies (this is addressed below).

---

[10]  A similar analysis employing dynamic conditional correlation methods, compiling hedge ratios using conditional variances and covariances in line with the relations presented in Appendix A, did not in fact lead to significantly different hedge effectiveness ratios. In addition, the combined use of the methods of Gibson et al. (2017) and a stochastic volatility modelling approach designed to address the presence of isolated outliers, due to Chan and Grant (2016), also failed to change the conclusions drawn from the more straightforward application of a rolling linear regression approach. These alternative methods may however lead to improvements when used at higher frequency with regular updating. Extending the analysis to such a frequency is beyond the scope or needs of the present study.

[11]  The cases of Austria (AT) and Finland (FI) are not shown above but are quite similar to the case of NL.

**Table 1.** Hedge effectiveness: Previous to the Sovereign Debt Crisis.

| | **1 SBBS** | | | **2-SBBS** | | | **3-SBBS** |
|---|---|---|---|---|---|---|---|
| **Hedge =** | **Snr** | **Mezz** | **Jnr** | **Snr-Mezz** | **Snr-Jnr** | **Mezz-Jnr** | **Snr-Mezz-Jnr** |
| AT(i) | −62 | −61 | −35 | −67 | −70 | −50 | −72 |
| AT(ii) | −73 | −72 | −35 | −77 | −82 | −57 | −84 |
| BE(i) | −65 | −63 | −36 | −72 | −75 | −52 | −77 |
| BE(ii) | −71 | −70 | −37 | −76 | −80 | −58 | −83 |
| DE(i) | −79 | −78 | −32 | −84 | −84 | −46 | −87 |
| DE(ii) | −85 | −81 | −31 | −86 | −88 | −49 | −89 |
| ES(i) | −55 | −55 | −36 | −62 | −66 | −53 | −69 |
| ES(ii) | −62 | −61 | −36 | −69 | −73 | −58 | −75 |
| FI(i) | −70 | −69 | −35 | −72 | −75 | −46 | −76 |
| FI(ii) | −79 | −77 | −36 | −81 | −84 | −53 | −84 |
| FR(i) | −72 | −71 | −37 | −78 | −80 | −53 | −83 |
| FR(ii) | −76 | −75 | −37 | −81 | −84 | −59 | −88 |
| GR(i) | −36 | −33 | −27 | −46 | −51 | −49 | −55 |
| GR(ii) | −46 | −44 | −33 | −60 | −60 | −58 | −67 |
| IE(i) | −42 | −40 | −26 | −47 | −51 | −39 | −52 |
| IE(ii) | −66 | −62 | −33 | −70 | −72 | −52 | −72 |
| IT(i) | −50 | −47 | −35 | −63 | −65 | −59 | −72 |
| IT(ii) | −56 | −50 | −37 | −69 | −70 | −64 | −77 |
| NL(i) | −69 | −68 | −37 | −75 | −78 | −54 | −81 |
| NL(ii) | −77 | −75 | −36 | −80 | −83 | −58 | −86 |
| PT(i) | −50 | −48 | −34 | −59 | −63 | −54 | −67 |
| PT(ii) | −62 | −60 | −38 | −69 | −73 | −61 | −77 |

Note: Using two different metrics, this table shows the percentage change in risk exposures achieved by hedging a position in a particular sovereign bond using a position in one or more tranches of a 70:20:10 Sovereign Bond Backed Securitisation in the pre-SDC period (this is used as an indicator of *hedge effectiveness* in the text). Issuers of the bonds are indicated at the beginning of each row as follows; AT (Austria), BE (Belgium), DE (Germany), ES (Spain), FI (Finland), FR (France), GR (Greece), IE (Ireland), IT (Italy), NL (Netherlands) and PT (Portugal). The first row for each case contains the percentage change in risk due to hedging where risk is measured as *standard deviation*. The second row for each case contains the percentage change in risk achieved through hedging where risk is measured as the range between the 5th and 95th quantiles of the returns distribution (implying VaR bounds). Columns of results are arranged in three broad categories. The first category concerns the use of a single SBBS tranche in the hedge. The second category concerns the use of a pair of SBBS in hedging. The last category involves the use of all three SBBS in the hedge. The most effective hedge instrument/combination within the first two categories is highlighted with colour.

**Table 2.** Hedge effectiveness Sov-debt-crisis.

| | **1 SBBS** | | | **2-SBBS** | | | **3-SBBS** |
|---|---|---|---|---|---|---|---|
| **Hedge =** | **Snr** | **Mezz** | **Jnr** | **Snr-Mezz** | **Snr-Jnr** | **Mezz-Jnr** | **Snr-Mezz-Jnr** |
| AT(i) | −24 | −11 | 0 | −32 | −16 | 4 | −26 |
| AT(ii) | −32 | −19 | −2 | −41 | −39 | −5 | −41 |
| BE(i) | −3 | −4 | −2 | −27 | 10 | −16 | −20 |
| BE(ii) | −2 | −2 | 0 | −27 | −10 | −17 | −29 |
| DE(i) | −68 | 0 | 7 | −72 | −67 | 4 | −71 |
| DE(ii) | −69 | 4 | 5 | −73 | −69 | −5 | −73 |
| ES(i) | 1 | 10 | 1 | −33 | 10 | −31 | −28 |
| ES(ii) | −3 | 15 | 5 | −29 | −13 | −34 | −35 |
| FI(i) | −52 | −7 | 3 | −52 | −49 | 6 | −47 |
| FI(ii) | −54 | −4 | 2 | −54 | −54 | 4 | −55 |
| FR(i) | −23 | −12 | 0 | −35 | −15 | 0 | −31 |
| FR(ii) | −30 | −12 | 2 | −38 | −32 | 2 | −38 |
| GR(i) | 0 | 1 | 0 | 0 | −15 | −15 | −17 |
| GR(ii) | −4 | 13 | 11 | 2 | 26 | 28 | 23 |
| IE(i) | 2 | 7 | 2 | −3 | 1 | −2 | 1 |
| IE(ii) | −1 | 6 | 3 | −5 | −8 | −7 | −6 |

**Table 2.** *Cont.*

| Hedge = | 1 SBBS | | | 2-SBBS | | | 3-SBBS |
|---|---|---|---|---|---|---|---|
| | Snr | Mezz | Jnr | Snr-Mezz | Snr-Jnr | Mezz-Jnr | Snr-Mezz-Jnr |
| IT(i) | 0 | 10 | 1 | −44 | 18 | −39 | −37 |
| IT(ii) | 2 | 13 | 3 | −40 | −9 | −43 | −44 |
| NL(i) | −49 | −9 | 2 | −48 | −46 | 7 | −43 |
| NL(ii) | −53 | −6 | 5 | −52 | −52 | 3 | −51 |
| PT(i) | 1 | 5 | 1 | −1 | 1 | −2 | 0 |
| PT(ii) | 1 | 2 | 1 | −5 | −10 | −8 | −9 |

Note: Using two different metrics, this table shows the percentage change in risk exposures achieved by hedging a position in a particular sovereign bond using a position in one or more tranches of a 70:20:10 Sovereign Bond Backed Securitisation during the SDC period (this is used as an indicator of *hedge effectiveness* in the text). Issuers of the bonds are indicated at the beginning of each row as follows; AT (Austria), BE (Belgium), DE (Germany), ES (Spain), FI (Finland), FR (France), GR (Greece), IE (Ireland), IT (Italy), NL (Netherlands) and PT (Portugal). The first row for each case contains the percentage change in risk due to hedging where risk is measured as *standard deviation*. The second row for each case contains the percentage change in risk achieved through hedging where risk is measured as the range between the 5th and 95th quantiles of the returns distribution (implying VaR bounds). Columns of results are arranged in three broad categories. The first category concerns the use of a single SBBS tranche in the hedge. The second category concerns the use of a pair of SBBS in hedging. The last category involves the use of all three SBBS in the hedge. The most effective hedge instrument/combination within the first two categories is highlighted with colour.

**Table 3.** Hedge effectiveness Post-Sov-debt-crisis.

| Hedge = | 1 SBBS | | | 2-SBBS | | | 3-SBBS |
|---|---|---|---|---|---|---|---|
| | Snr | Mezz | Jnr | Snr-Mezz | Snr-Jnr | Mezz-Jnr | Snr-Mezz-Jnr |
| AT(i) | −45 | −22 | 0 | −47 | −49 | −10 | −49 |
| AT(ii) | −51 | −25 | 0 | −53 | −57 | −14 | −56 |
| BE(i) | −44 | −26 | −2 | −48 | −53 | −13 | −52 |
| BE(ii) | −50 | −28 | −3 | −53 | −57 | −15 | −57 |
| DE(i) | −73 | −13 | 4 | −74 | −73 | −8 | −75 |
| DE(ii) | −72 | −10 | 4 | −73 | −73 | −7 | −74 |
| ES(i) | −2 | 2 | −3 | −32 | −26 | −42 | −43 |
| ES(ii) | −4 | −6 | −4 | −29 | −28 | −41 | −43 |
| FI(i) | −52 | −16 | 1 | −53 | −55 | −9 | −55 |
| FI(ii) | −59 | −18 | 1 | −60 | −62 | −11 | −62 |
| FR(i) | −50 | −27 | −2 | −55 | −58 | −15 | −59 |
| FR(ii) | −54 | −28 | −2 | −56 | −61 | −16 | −61 |
| GR(i) | 0 | 7 | 7 | −8 | −8 | 2 | −8 |
| GR(ii) | 5 | 6 | 8 | 3 | 11 | 17 | 12 |
| IE(i) | −10 | −11 | −3 | −21 | −22 | −19 | −27 |
| IE(ii) | −14 | −17 | −5 | −29 | −28 | −23 | −35 |
| IT(i) | −3 | 1 | −4 | −41 | −28 | −50 | −53 |
| IT(ii) | −7 | −5 | −4 | −41 | −34 | −52 | −54 |
| NL(i) | −53 | −18 | 1 | −54 | −56 | −9 | −56 |
| NL(ii) | −60 | −18 | 0 | −61 | −64 | −11 | −65 |
| PT(i) | 0 | 2 | 0 | −13 | −15 | −21 | −21 |
| PT(ii) | −1 | 2 | 0 | −13 | −17 | −25 | −26 |

Note: Using two different metrics, this table shows the percentage change in risk exposures achieved by hedging a position in a particular sovereign bond using a position in one or more tranches of a 70:20:10 Sovereign Bond Backed Securitisation in the post-SDC period (this is used as an indicator of *hedge effectiveness* in the text). Issuers of the bonds are indicated at the beginning of each row as follows; AT (Austria), BE (Belgium), DE (Germany), ES (Spain), FI (Finland), FR (France), GR (Greece), IE (Ireland), IT (Italy), NL (Netherlands) and PT (Portugal). The first row for each case contains the percentage change in risk due to hedging where risk is measured as *standard deviation*. The second row for each case contains the percentage change in risk achieved through hedging where risk is measured as the range between the 5th and 95th quantiles of the returns distribution (implying VaR bounds). Columns of results are arranged in three broad categories. The first category concerns the use of a single SBBS tranche in the hedge. The second category concerns the use of a pair of SBBS in hedging. The last category involves the use of all three SBBS in the hedge. The most effective hedge instrument/combination within the first two categories is highlighted with colour.

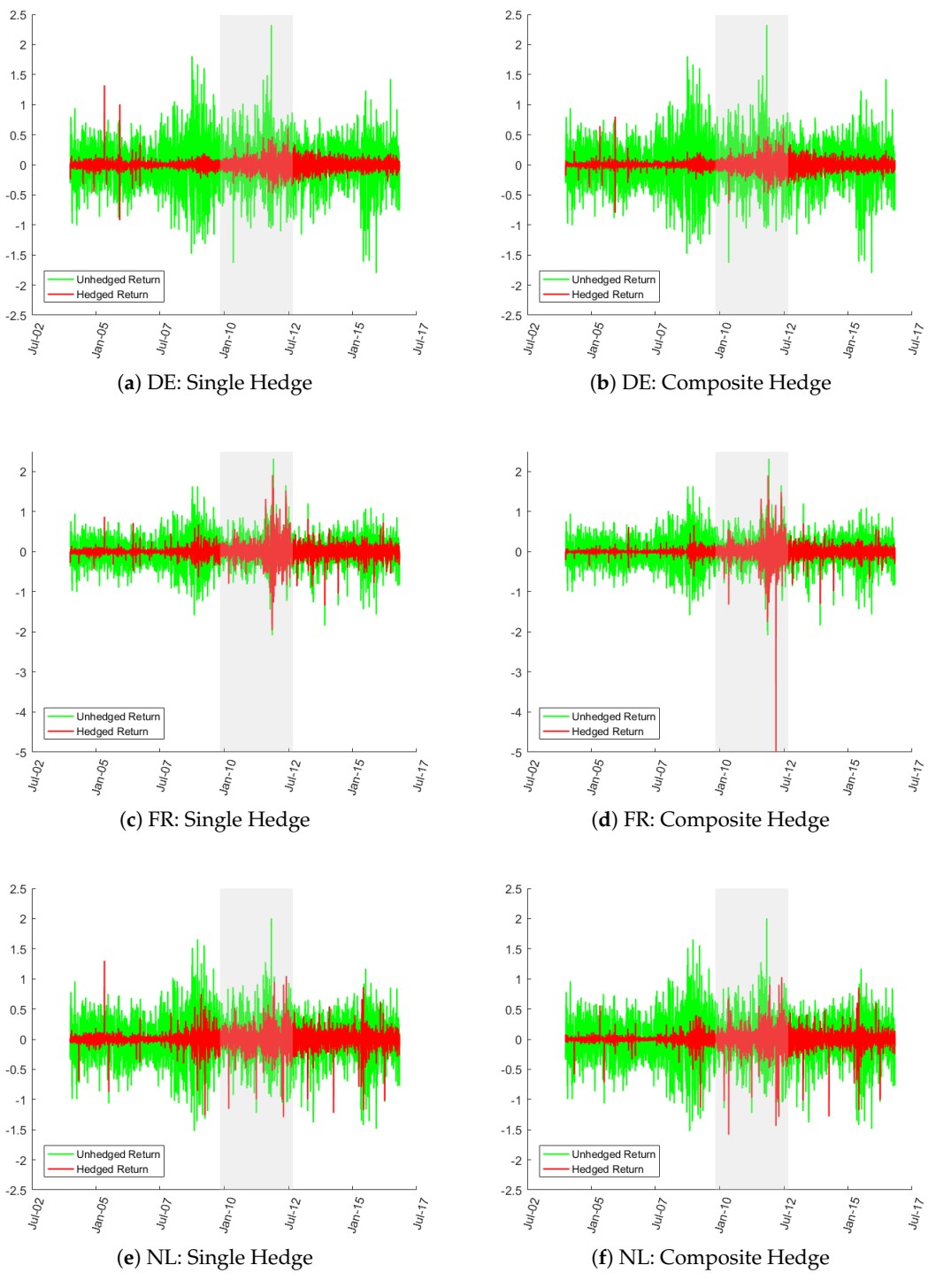

**Figure 3.** Single and Composite Hedging (DE, FR, NL). Note: This figure facilitates a comparison of the dispersion of returns on hedged long positions in German, French and Dutch sovereign bonds respectively with the dispersion of returns on the same sovereign bonds without hedging. The first column of figures concern the case of hedging the long positions using only the senior SBBS. The second column of figures concerns the case of hedging the individual bond positions with both the senior and mezzanine SBBS. The returns are measured in basis points (left axis) and the bonds considered are those with a 10 year term-to-maturity. The cases of Austria (AT) and Finland (FI) are not shown above but are quite similar to the case of NL.

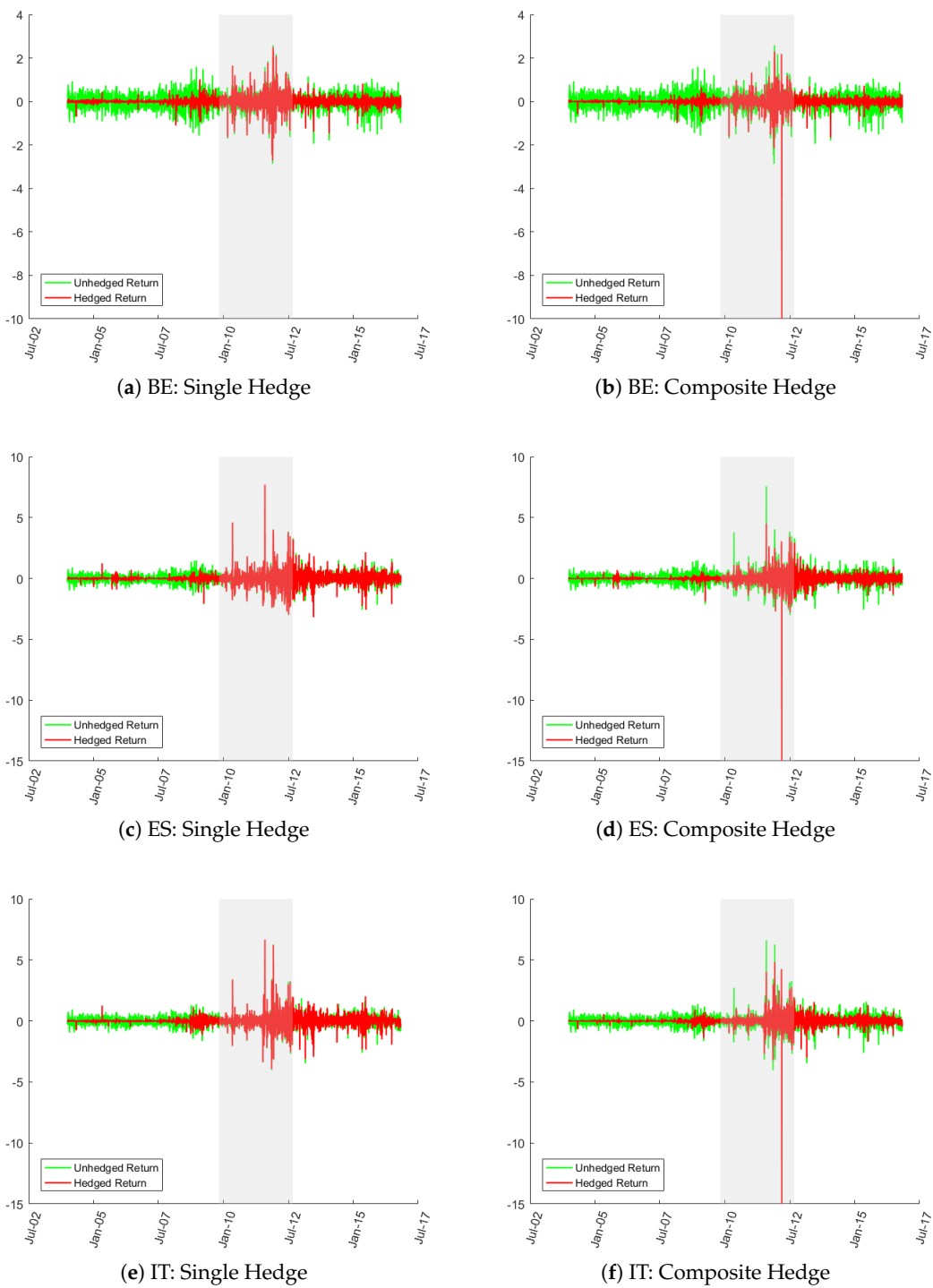

**Figure 4.** Single and Composite Hedging (BE, ES, IT). Note: This figure facilitates a comparison of the dispersion of returns on hedged long positions in Belgian, Spanish and Italian sovereign bonds respectively with the dispersion of returns on the same sovereign bonds without hedging. The first column of figures concern the case of hedging the long positions using only the senior SBBS. The second column of figures concerns the case of hedging the individual bond positions with both the senior and mezzanine SBBS. The returns are measured in basis points (left axis) and the bonds considered are those with a 10 year term-to-maturity.

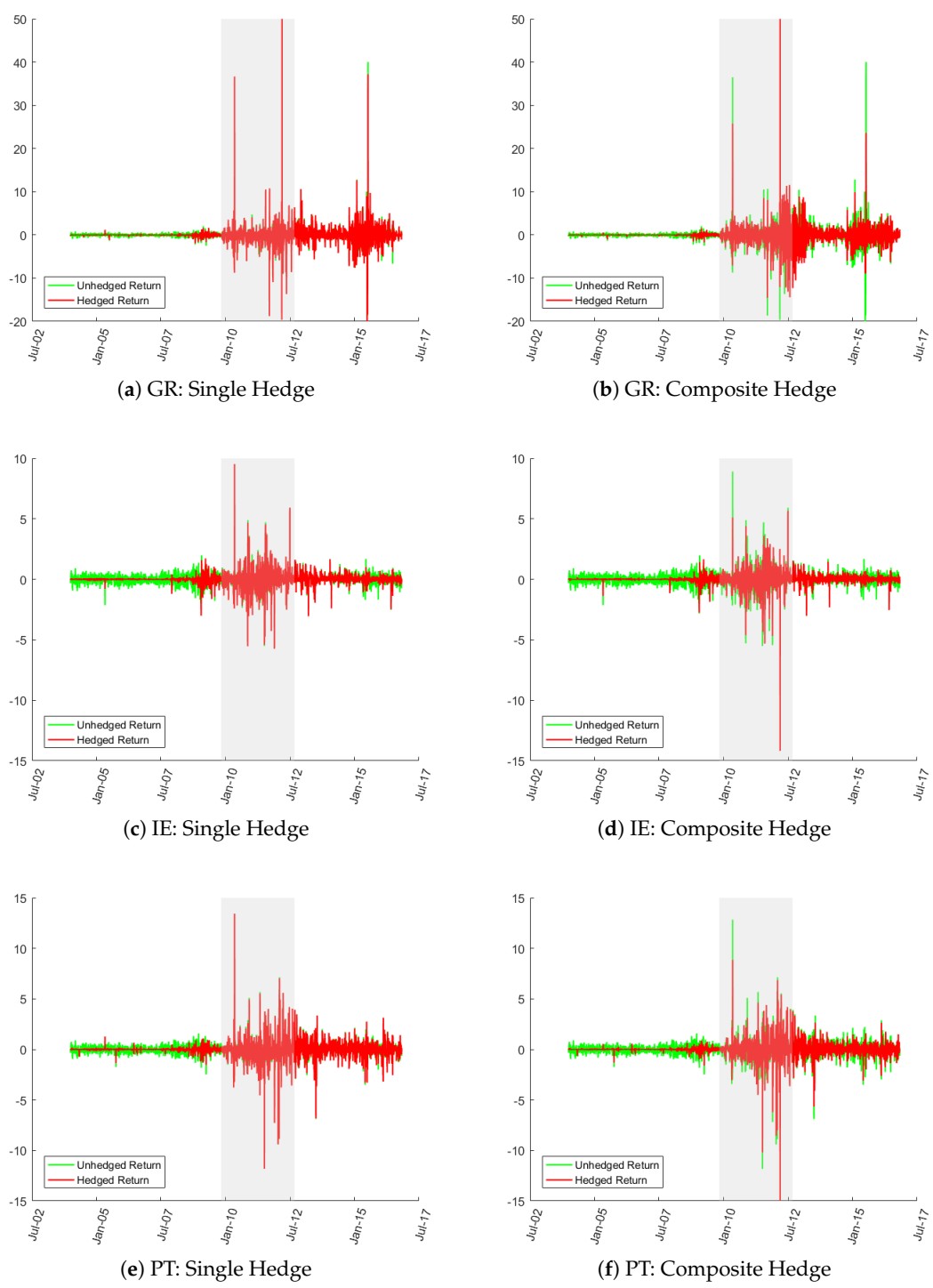

**Figure 5.** Single and Composite Hedging (GR, IE, PT). Note: This figure facilitates a comparison of the dispersion of returns on hedged long positions in Greek, Irish and Portuguese sovereign bonds respectively with the dispersion of returns on the same sovereign bonds without hedging. The first column of figures concern the case of hedging the long positions using only the senior SBBS. The second column of figures concerns the case of hedging the individual bond positions with both the senior and mezzanine SBBS. The returns are measured in basis points (left axis) and the bonds considered are those with a 10 year term-to-maturity.

The tabulations of risk reductions can provide some more depth to the graphical analysis. The pre-Sovereign Debt Crisis period comparisons are in Table 1 and this is clearly a period where hedge effectiveness is high for all countries (at least for some of the seven hedges). The best hedges for each sovereign are highlighted in colour for the cases of 1-SBBS and 2-SBBS as hedges. In the case of the single hedge, it is the senior-SBBS that gives the best protection (the percentage reduction for the best single hedges—based on standard deviation of daily returns—ranges from 79% for DE to 36% for GR). In almost all cases, the 2-SBBS hedge provides some marginal improvement in hedge effectiveness over the 1-SBBS hedge case (all percentage reductions above 50%). In many cases the 3-SBBS hedge is best overall but this may not always be worthwhile from a cost and operational perspective.

Table 2 shows summary statistics for comparisons of risk changes through hedging during the sovereign debt crisis period. For the 1-SBBS hedge, only DE remains well hedged using the senior SBBS (reducing risk by 68%). Roughly half of the risk is avoided by single-SBBS hedging for the case of FI and NL while the remaining sovereigns are clearly not amenable to single-SBBS hedging in this crisis period. Moving to 2-SBBS or 3-SBBS hedging generally gives rise to some small risk reduction for most sovereigns relative to the single-SBBS case. Table 3 covers the recovery period (from Mario Draghi's London Speech until the end of the last quarter of 2016). By use of composite hedging, it is usually possible to reduce risks by half or more. The exceptions remain GR and PT.

*5.2. Post-Hedging Diversification of Risks*

Once risks have been hedged, there is potential for dealers to diversify remaining risks by operating simultaneously across many of the sovereign markets. At each moment of a trading day (and at the end of each day), it is likely that a dealer will have a portfolio of outstanding positions in many markets. The individual net hedged-positions are likely to be much smaller than unhedged positions (e.g., if the hedge-ratio is close to 1 then the individual net hedged-positions will be minor). These net positions in individual sovereigns could be subject to further netting (i.e., if some are net-long and others are net-short). Risk reduction from this additional type of netting is not included in the following analysis (i.e., portfolios will be restricted to be hedged long-positions in the components). This most probably implies a significant underestimate of the risk reduction that could be achieved by diversification.

The upper panel of Figure 6 compares returns on a cross-country portfolio of hedged and unhedged positions for the pre-SDC period. In this case, it is assumed that the portfolio results from an equal weighting on the underlying sovereigns. Firstly, it is clear that the portfolio of hedged positions has a much smaller dispersion than the portfolio of unhedged components. Despite the equal weighting of components and the equal capital exposure assumption, diversification reduces risks much more for the portfolio with hedged positions and this is due to the fact that there are mostly idiosyncratic risks surviving in the hedged case while the unhedged case will involve considerable systematic risk.

The lower panel of Figure 6 concerns the more volatile conditions of the SDC and the recovery. In this case, there is not much benefit from the hedging because most risk is actually idiosyncratic. This means that diversification is equally effective in the case of the unhedged portfolio. In general, the risks are much lower than risks for typical single sovereigns during this period. Hedging starts to matter again during the recovery.

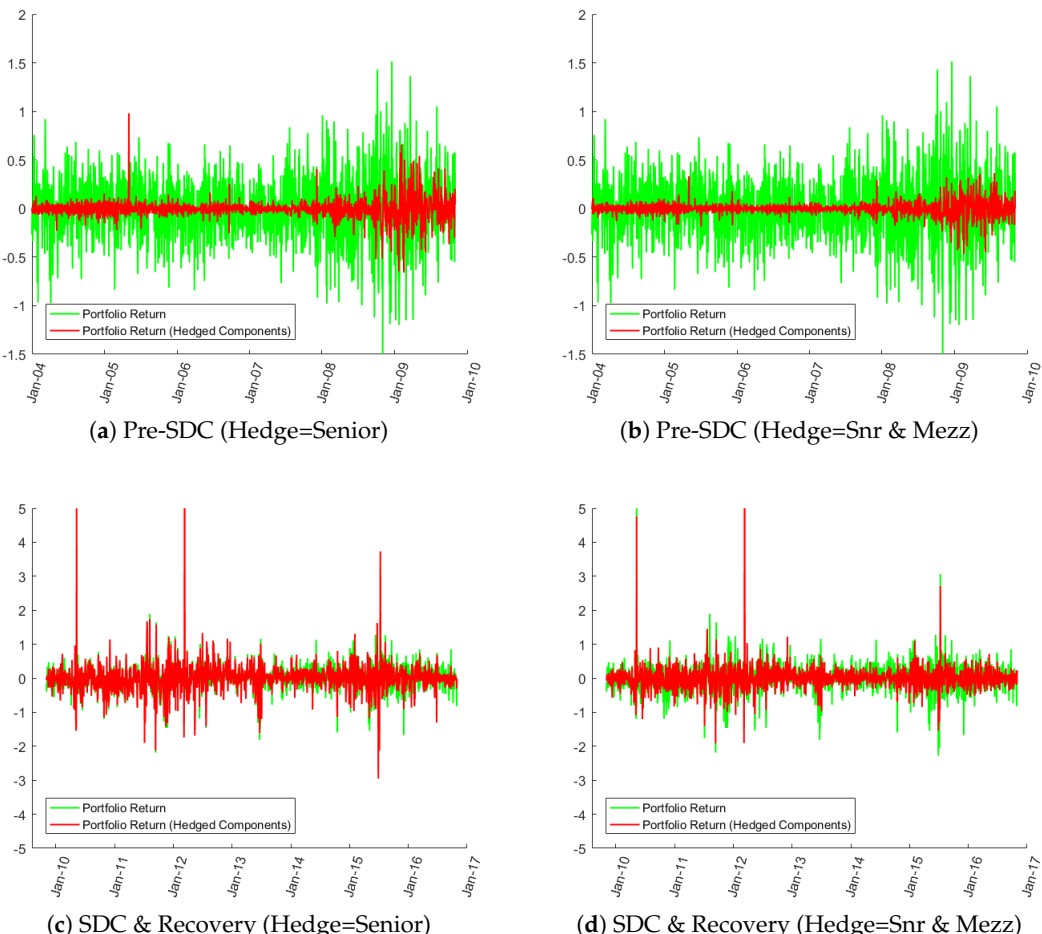

**Figure 6.** Portfolio returns with and without hedged components. Note: This figure facilitates a comparison of the dispersion of returns on a portfolio of hedged long positions in 11 sovereigns with the dispersion of returns on the same portfolio of long positions without hedging. The first column of figures concern the case of hedging the components of the portfolio of long positions using only the senior SBBS. The second column of figures concerns the case of hedging the individual bond positions with both the senior and mezzanine SBBS. The returns are measured in basis points (left axis) and the bonds considered are those with a 10 year term-to-maturity. The returns are measured in basis points (left axis).

A related comparison is that of the dispersion of returns on the safest single sovereign asset and those of the portfolio of hedged sovereign positions (the German 10 year Bund returns are used for this discussion, but these are almost indistinguishable from the returns on the similarly safe senior SBBS). Figure 7 displays these returns through time for the pre-SDC period and the sample from the start of the SDC through to the more recent recovery. It is very obvious that the portfolio of hedged sovereigns has an extremely low risk level when compared with the most safe sovereign (especially in the pre-SDC period). This demonstrates the effectiveness of diversification in reducing risks when risks are idiosyncratic and not too extreme. The effectiveness of diversification is compromised from the start of the SDC, but there is still some significant reduction in risk for the majority of the sample (with a small number of extreme outliers). For normal risk levels, diversification is a strong driver of risk reduction.

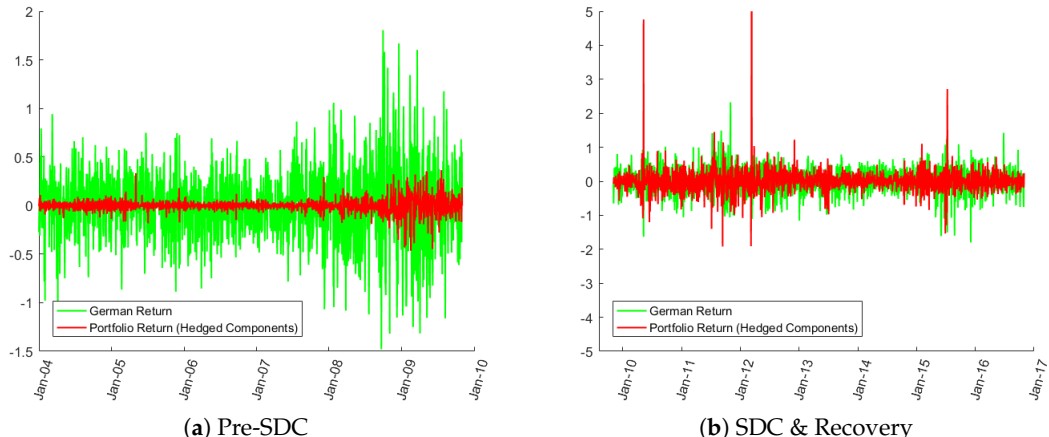

**Figure 7.** German 10-year returns compared with those from an 11-country portfolio of sovereigns hedged with SBBS. Note: This figure facilitates a comparison of the dispersion of returns on a portfolio of hedged long positions in 11 sovereigns with the dispersion of returns on a single sovereign bond that is widely considered to be the safest sovereign bond investment in the euro area (namely the German Bund). The long positions are hedged using the senior SBBS. The returns are measured in basis points (left axis) and the bonds considered are those with a 10 year term-to-maturity. (**a**) shows the comparison for the period previous to the Sovereign Debt Crisis while (**b**) covers the period of the Sovereign Debt Crisis and recovery. The hedged portfolio has returns with a dispersion which is much lower than the safe haven asset in the non-crisis sub-sample. Even in the crisis and recover periods, the hedged portfolio compares favourably with the safe asset investment in terms of dispersion of returns.

Table 4 broadens the analysis of the relative risks following diversification to the case of a size-weighted portfolio and to other maturities. These results pertain to the case where the senior SBBS is used as the hedge instrument.[12] The tabulated results for the 10-year bonds confirm what was clear from the graphical analysis. In the pre-SDC period, most risk measures indicate a reduction of around 70% in risk due to a combination of diversification and hedging relative to diversification on its own. It also confirms that risks after hedging and diversifying are less than a third of those faced by investing in the lowest risk sovereign (close to what is the maximum reduction possible when investing the same capital in 11 assets with similar idiosyncratic risks rather than 1). In this period (for the 10 year term), there is little difference between the equal and size-weighted portfolio results.

For the 10 year term, there is little benefit from hedging before diversifying during the Sovereign Debt Crisis period (the ratio of the risks on portfolio returns when components are hedged is roughly equal to the risks with unhedged components which is tantamount to saying that almost all risks are idiosyncratic). Diversification reduces risks again in the Recovery period and this is particularly true for the size-weighted portfolio. Risk is relatively high for the equally-weighted portfolio compared with the German bond for the Sovereign Debt Crisis period, but otherwise risk is similar to or less than that of the German for the SDC and Recovery, respectively.

In the case of the 5-year term, risks are reduced by around 45% in the pre-SDC period via a combination of hedging and diversification relative to just diversification (the extra risk reduction achieved in the 2 year case is similar to that of the 10 year). As with the 10 year case, hedging does not achieve significant extra benefit in terms of risk reduction for the 5-year bonds during the SDC (and this is also true of the two year term-to-maturity). There is about a ten percentage point difference in the risk reduction due to hedging during the Recovery between the 2 and 5 year terms with more

---

[12] The weights used in the analysis are related to the size of the individual sovereign float relative to that of the total of 11 sovereigns in the euro area market (weights are provided in the notes for Table 4).

reduction achieved in the 5 year category. As for comparison with the German sovereign, an equally weighted portfolio of sovereigns has risk of 1.52 and 3.52 times the volatility of the German bond (for the 5 and 2 year case, respectively) during the SDC (the associated relative VaR measures and the size-weighted comparisons show a smaller difference in these risks, but, in all cases, risks are greater than holding just the German). The Recovery period does not appear to be as positive for risk reduction through the combination of hedging and diversification for the 2 year term. However, portfolio risks remain low and are not much greater than those of the lowest risk sovereign.

**Table 4.** Diversification and hedge effectiveness.

| | Pre-Crisis | | Sov Debt Crisis | | Recovery | |
|---|---|---|---|---|---|---|
| Weighting = | Equal | Size | Equal | Size | Equal | Size |
| 10-Year | | | | | | |
| EA(Hedged)/EA (i) | −68 | −71 | 0 | −3 | −12 | −24 |
| EA(Hedged)/EA (ii) | −73 | −74 | −4 | −7 | −18 | −31 |
| EA(Hedged)/DE (i) | −71 | −73 | 8 | −1 | −4 | −24 |
| EA(Hedged)/DE (ii) | −76 | −76 | −2 | −14 | −11 | −29 |
| 5-Year | | | | | | |
| EA(Hedged)/EA (i) | −47 | −43 | 1 | −1 | −17 | −15 |
| EA(Hedged)/EA (ii) | −46 | −44 | 0 | 1 | −20 | −21 |
| EA(Hedged)/DE (i) | −50 | −46 | 52 | 25 | −21 | −3 |
| EA(Hedged)/DE (ii) | −50 | −46 | 34 | 12 | −27 | −10 |
| 2-Year | | | | | | |
| EA(Hedged)/EA (i) | −62 | −64 | 1 | −1 | −5 | −5 |
| EA(Hedged)/EA (ii) | −69 | −72 | 1 | 4 | −7 | −10 |
| EA(Hedged)/DE (i) | −70 | −67 | 252 | 81 | 21 | 47 |
| EA(Hedged)/DE (ii) | −76 | −74 | 186 | 64 | 15 | 38 |

Note: Using two different metrics, this table shows the percentage change in risk exposures achieved by hedging and diversifying (using the optimal hedge position in one or more tranches of a 70:20:10 Sovereign Bond Backed Securitisation). The portfolio of assets being hedged and diversified involves the 10 year bonds issued by; AT (Austria), BE (Belgium), DE (Germany), ES (Spain), FI (Finland), FR (France), GR (Greece), IE (Ireland), IT (Italy), NL (Netherlands) and PT (Portugal). The first row for each case contains the percentage change in risk due to hedging where risk is measured as *standard deviation*. The second row for each case contains the percentage change in risk achieved through hedging where risk is measured as the range between the 5th and 95th quantiles of the returns distribution (implying VaR bounds). The risk reduction is measured relative to the unhedged portfolio and relative to an investment in just the German Bund which is considered the safe haven asset. Columns of results permit a comparison across three sub-sample periods and for risk reductions achieved by diversifying across an equally-weighted and size-weighted portfolio, respectively.

*5.3. Summary of Results for Extensions*

The possibility that similar hedge (and diversification) strategies might be just as good as SBBS has been ignored. The effectiveness of hedging using futures (in particular, German Bund and Italian BTP bond futures) is addressed in Appendix B. The principal finding here is that the senior SBBS is a much better hedge than Bund futures (even ignoring the fact that hedging with futures involves other additional costs and basis risks). Additionally, where SBBS are ineffective as hedges for smaller sovereigns (such as IE and PT), the futures are also very ineffective so there is no clear superiority in terms of hedge effectiveness using futures rather than SBBS.

Appendix B also contains details of the generalisation of the analysis to other maturities and to the case where SBBS yields are generated using a multivariate t-copula to trigger defaults. The outcome of these extensions is straightforward to summarise. Firstly, extending to shorter maturities leads to a general reduction in hedge effectiveness on average in the pre-crisis period and a less pervasive reduction in hedge effectiveness for the two-year maturity in the recovery period. Ultimately, this does not impact greatly on the risks that remain after diversification. Secondly, since the t-copula produces a higher incidence of extreme losses that more often spillover to mezzanine and senior tranches, the yield movements that result are less correlated with those of the German sovereign bonds than is the case with the Gaussian copula. The senior tranche therefore becomes less effective as a hedge for German bonds. Otherwise, the changes in hedge effectiveness due to use of the t-copula are insignificant.

## 6. Conclusions

This paper assesses how the existence of sovereign bond-backed securities affects dealer intermediation in individual euro area sovereigns. An arbitrage relation is used to demonstrate that the hedging of inventory risks with SBBS eliminates most systematic inventory holding risks. Furthermore, it is asserted that diversification of intermediation activities across countries, in combination with hedging, then produces further reductions in exposures. These assertions are tested using estimated SBBS yields and the findings are generally positive. Even if one assumes a similar capital exposure under hedging and not-hedging, there is still a marked reduction in risks through hedging and diversification. Overall, assuming regulation does not excessively penalise netting of inventory positions in different sovereign and SBBS, and if SBBS are sufficiently liquid, a significant improvement in trading costs across all European sovereign debt markets seems plausible.

Risk reduction through diversification (after hedging) was assessed using equal-weighted and size-weighted portfolios of long positions in the bonds from the underlying sovereigns. This could be misleading since it is quite likely that actual inventories would be some mixture of long and short positions. This simplification will tend to underestimate the risk-reducing benefits of diversification. Since holding period investment returns in sovereign markets are predominantly positively correlated (except when there is a pronounced flight to safety), a combination of long and short positions will lead to more netting of valuation changes and this will most likely reduce risk relative to what is reported for the present analysis. This represents a fruitful avenue for future investigation.

A further useful extension of the above analysis would be to assess whether there are liquidity spillovers among the SBBS tranches. In fact, these spillovers have already been identified in terms of the incremental hedge effectiveness that was achieved when portfolios that included all three SBBS were analysed. However, in the developmental phase of the SBBS market, liquidity spillovers between the SBBS themselves could be important for liquidity of the junior tranche which will have the smallest issuance volume among the three. It seems likely that the senior SBBS would be helpful to dealers in providing quotes in the mezzanine market (purely via hedging). A spillover from the mezzanine to the junior is then possible. Preliminary analysis of the correlations between the tranches under most circumstances suggests that these hedging avenues for liquidity spillovers would be positive. Of course, in crisis circumstances, flight to safety could disrupt correlations more dramatically than can be anticipated by dealers and this would undermine the reliability of risk reduction strategies based purely on hedging one SBBS with another. This is worthy of further investigation.

It is important to acknowledge that providing liquidity by relying on a parallel market for hedging requires adequate funding liquidity, regulation that permits the netting of inventory positions and broad-based diversification by dealers. Any systemic contraction in availability of funding liquidity is likely to disproportionately affect liquidity in markets that depend on hedging (see Brunnermeier and Pedersen 2009). It is important therefore that dealers have adequate capital to withstand relatively large shocks.[13] Regulation that prevents netting of positions of intermediaries would constrain positive liquidity spillover effects. Similarly, large changes in correlations can magnify risks during a crisis so trading systems need to be dynamically flexible and capable of managing such complexity. This may increase operational costs.

One important policy implication of the analysis above is that primary dealers will seek to have a presence in more markets. This follows from the fact that much of the risk at local level cannot be hedged using SBBS but is easily diversified. Dealers with, on average, a more diversified portfolio will face lower risks and will be able to out-price non-diversified dealers. Most markets in Europe have a majority of dealers with diversified market making activities, but some additional diversification is

---

[13] It has been shown by Baranova et al. (2017) that recent tightening of capital and leverage requirements of financial intermediaries has damaged liquidity provision during calm markets' conditions, but it has helped to protect liquidity during crises' circumstances. Getting the balance right is therefore crucial.

likely to occur. There is obviously a trade-off here between specialisation, which helps to make price discovery more efficient, and diversification for risk management purposes.

**Funding:** This research received no external funding.

**Acknowledgments:** I am grateful to Sam Langfield, Marco Pagano, Richard Portes, Philip Lane, Javier Suárez, Kitty Moloney, Patrick Haran, Angelo Ranaldo, Thomas Reininger, Martin Phul and members of the ESRB High-Level Task Force on Safe Assets for helpful comments. Views expressed are those of the author and do not necessarily represent the views of the task force. Technical assistance from Paul Huxley at Matlab is also gratefully acknowledged.

**Conflicts of Interest:** The author declares no conflict of interest.

## Appendix A. Optimal Hedging

The optimal hedge ratio is the one that minimises the variance of the hedged portfolio's return. To hedge quantity $Q_1$ of an asset using quantities of representative combinations of hedge instruments denoted $Q_2$, $Q_3$ and $Q_4$ gives the following hedge ratios (where $\rho_{ij}$ is the correlation between returns on assets $i$ and $j$, and $\sigma_i$ is the standard deviation of returns on asset $i$).

**One hedge instrument:** $\frac{Q_2}{Q_1} = -\frac{\rho_{12}\sigma_{R_1}\sigma_{R_2}}{\sigma_{R_2}^2}$,

**Two hedge instruments:** $\frac{Q_2}{Q_1} = -\frac{(\rho_{12}-\rho_{13}\rho_{23})\sigma_1\sigma_2}{(1-\rho_{23}^2)\sigma_2^2}$ and $\frac{Q_3}{Q_1} = -\frac{(\rho_{13}-\rho_{12}\rho_{23})\sigma_1\sigma_3}{(1-\rho_{23}^2)\sigma_3^2}$,

**Three hedge instruments:**

$$\frac{Q_2}{Q_1} = -\frac{(1-\rho_{34}^2)\rho_{12} - (\rho_{23}-\rho_{24}\rho_{34})\rho_{13} + (\rho_{23}\rho_{34}-\rho_{24})\rho_{14}}{(1-\rho_{23}^2-\rho_{24}^2-\rho_{34}^2+2\rho_{23}\rho_{24}\rho_{34})\sigma_2},$$

$$\frac{Q_3}{Q_1} = -\frac{(1-\rho_{24}^2)\rho_{13} - (\rho_{23}-\rho_{24}\rho_{34})\rho_{12} + (\rho_{23}\rho_{24}-\rho_{34})\rho_{14}}{(1-\rho_{23}^2-\rho_{24}^2-\rho_{34}^2+2\rho_{23}\rho_{24}\rho_{34})\sigma_3},$$

$$\frac{Q_4}{Q_1} = -\frac{(1-\rho_{23}^2)\rho_{14} - (\rho_{34}-\rho_{24}\rho_{23})\rho_{13} + (\rho_{23}\rho_{34}-\rho_{24})\rho_{12}}{(1-\rho_{23}^2-\rho_{24}^2-\rho_{34}^2+2\rho_{23}\rho_{24}\rho_{34})\sigma_4}.$$

Correlations used in the the above calculations can be derived from dynamic estimates of variances as described in Gibson et al. (2017). For example, since the variance of (X + Y) = variance(X) + Variance(Y) + 2 Covariance(X,Y) the Covariance can be constructed from a rearrangement of estimates of variance(X + Y), variance(X) and Variance(Y).

## Appendix B. Robustness

*Appendix B.1. Comparing with Futures as Hedge*

The comparison of hedge effectiveness using SBBS versus BTP and/or Bund futures can be considered by reference to Table A1. The first notable outcome is that the senior SBBS is better than the Bund future even in the case of hedging German 10-year bond risk. The returns on a portfolio of Bund and an optimal hedge position in Bund futures has a standard deviation of around 67% and 64% respectively of the unhedged position in the SDC period and the recovery, respectively. The portfolio involving the senior SBBS by comparison has a relative standard deviation of 32% and 27% in these periods (the relative VaR in both periods is also much lower when the SBBS is used to hedge). Interestingly, the BTP future is a better hedge for the Italian bond than any of the SBBS individually, but a combination of SBBS achieves similar hedge effectiveness.

Overall, the Bund futures contract is a weaker hedge for most sovereign bond positions compared with SBBS. Where SBBS are ineffective as hedges for smaller sovereigns (such as IE and PT), the futures are also very ineffective. These sovereigns have a lot of idiosyncratic unhedgable risk, but their risks are significantly reduced through diversification.

**Table A1.** Futures-based hedge effectiveness SDC and Recovery.

| Hedge = | Sov Debt Crisis | | | Recovery | | |
|---|---|---|---|---|---|---|
| | BUND | BTP | BUND+BTP | BUND | BTP | BUND+BTP |
| AT(i) | −28 | 7 | −27 | −17 | −3 | −19 |
| AT(ii) | −26 | 13 | −24 | −23 | −7 | −29 |
| BE(i) | −1 | −11 | −19 | −10 | −6 | −16 |
| BE(ii) | 10 | 2 | −12 | −16 | −5 | −22 |
| DE(i) | −33 | 3 | −33 | −36 | −3 | −35 |
| DE(ii) | −32 | 6 | −32 | −44 | −4 | −44 |
| ES(i) | 2 | −36 | −33 | −2 | −24 | −23 |
| ES(ii) | −2 | −33 | −35 | −4 | −26 | −26 |
| FI(i) | −34 | 5 | −33 | −32 | −3 | −32 |
| FI(ii) | −38 | 0 | −38 | −41 | −7 | −44 |
| FR(i) | −22 | 6 | −21 | −16 | −4 | −19 |
| FR(ii) | −18 | 12 | −22 | −20 | −8 | −27 |
| GR(i) | 3 | 13 | 17 | 0 | 0 | 0 |
| GR(ii) | 3 | 18 | 19 | 3 | 0 | 2 |
| IE(i) | 0 | −5 | −6 | −1 | 0 | −1 |
| IE(ii) | −4 | −2 | −5 | −4 | −1 | −3 |
| IT(i) | 3 | −52 | −49 | −3 | −46 | −45 |
| IT(ii) | 7 | −57 | −59 | −5 | −59 | −58 |
| NL(i) | −33 | 5 | −32 | −29 | −2 | −29 |
| NL(ii) | −29 | 3 | −31 | −37 | −3 | −38 |
| PT(i) | 0 | −4 | −5 | 0 | −3 | −2 |
| PT(ii) | 3 | −8 | −5 | 1 | −5 | −6 |

Note: Using two different metrics, this table shows the percentage change in risk exposures achieved by hedging a position in a particular sovereign bond using a position in futures on Bunds, BTPs and a combination of both. Issuers of the bonds are indicated at the beginning of each row as follows; AT (Austria), BE (Belgium), DE (Germany), ES (Spain), FI (Finland), FR (France), GR (Greece), IE (Ireland), IT (Italy), NL (Netherlands) and PT (Portugal). The first row for each case contains the percentage change in risk due to hedging where risk is measured as *standard deviation*. The second row for each case contains the percentage change in risk achieved through hedging where risk is measured as the range between the 5th and 95th quantiles of the returns distribution (implying VaR bounds). Columns of results permit a comparison of the risk reductions for the different hedges across two sub-sample periods.

*Appendix B.2. Hedge Effectiveness at Other Maturities*

Table A2 presents a comparison of hedge effectiveness at the 10-year term-to-maturity with those at five and two years to maturity (for conciseness only the results for hedging with all three SBBS are shown). In the pre-crisis period hedge effectiveness tends to decline as term shortens. The standard deviation of returns on the hedged positions relative to that of the unhedged positions (Rows (i)) increases on average from 0.28 at the 10 year term to 0.36 and 0.49 at the 5- and 2-year terms respectively. The relative value-at-risk comparison (Rows (ii)) increase on average less dramatically than the relative standard deviations (from 0.20 to 0.25 and 0.36, respectively). The average increase in these ratios in the pre-crisis period is a reasonably good indication of what happens at the individual sovereign level (the case of Finland is somewhat of an outlier).

The middle segment of Table A2 depicts the hedge effectiveness measures across terms-to-maturity for the period of the Sovereign Debt Crisis. Here, it is not so clear that any significant change occurs across the different terms on average, but there is more heterogeneity across sovereigns. The average of the ratio of standard deviations of returns on the hedged position relative to the unhedged, shown in the second last row of the table, move from 0.71 to 0.67 and then to 0.72. Relative VaRs also remain quite flat, moving from 0.67 at the 10-year term to 0.68 and 0.73 at the 5- and 2-year terms, respectively. The declines in these ratios tend to be more acute for ES, IT and PT. Significant increases occur (particularly for the 2-year term) for BE, DE, FI and NL. The hedge effectiveness results for the Recovery period show similar levels on average of the two main risk ratios between the 5- and 10-year maturities. A slightly sharper rise occurs in these ratios for the 2-year term and this is mainly explained by the FR case and two of the smaller sovereign markets (BE and FI).

**Table A2.** Hedge effectiveness: 10-, 5- & 2-year terms-to-maturity

| Term = | Pre-Crisis | | | Sov Debt Crisis | | | Recovery | | |
|---|---|---|---|---|---|---|---|---|---|
| | 10 Year | 5 Year | 2 Year | 10 Year | 5 Year | 2 Year | 10 Year | 5 Year | 2 Year |
| AT(i) | −72 | −65 | −47 | −26 | −29 | −23 | −49 | −44 | −24 |
| AT(ii) | −84 | −80 | −58 | −41 | −40 | −30 | −56 | −51 | −24 |
| BE(i) | −77 | −67 | −56 | −20 | −23 | −15 | −52 | −35 | −15 |
| BE(ii) | −83 | −81 | −76 | −29 | −26 | −20 | −57 | −45 | −10 |
| DE(i) | −87 | −85 | −75 | −71 | −69 | −48 | −75 | −72 | −68 |
| DE(ii) | −89 | −88 | −82 | −73 | −69 | −55 | −74 | −73 | −67 |
| ES(i) | −69 | −67 | −62 | −28 | −44 | −42 | −43 | −50 | −43 |
| ES(ii) | −75 | −80 | −77 | −35 | −38 | −36 | −43 | −48 | −37 |
| FI(i) | −76 | −55 | −26 | −47 | −45 | −20 | −55 | −41 | −27 |
| FI(ii) | −84 | −80 | −30 | −55 | −56 | −26 | −62 | −56 | −24 |
| FR(i) | −83 | −81 | −75 | −31 | −33 | −29 | −59 | −49 | −31 |
| FR(ii) | −88 | −86 | −82 | −38 | −38 | −35 | −61 | −52 | −27 |
| GR(i) | −55 | −45 | −34 | −17 | 0 | 1 | −8 | | |
| GR(ii) | −67 | −60 | −61 | 23 | 31 | 42 | 12 | | |
| IE(i) | −52 | −40 | −17 | 1 | −13 | | −27 | −14 | |
| IE(ii) | −72 | −40 | −27 | −6 | −10 | | −35 | −19 | |
| IT(i) | −72 | −64 | −59 | −37 | −59 | −62 | −53 | −54 | −60 |
| IT(ii) | −77 | −70 | −73 | −44 | −51 | −61 | −54 | −55 | −57 |
| NL(i) | −81 | −71 | −54 | −43 | −42 | −25 | −56 | −47 | −39 |
| NL(ii) | −86 | −84 | −72 | −51 | −53 | −42 | −65 | −57 | −36 |
| PT(i) | −67 | −59 | −58 | 0 | −9 | −17 | −21 | −13 | −8 |
| PT(ii) | −77 | −76 | −69 | −9 | −5 | −3 | −26 | −11 | 1 |
| Avg(i) | −72 | −64 | −51 | −29 | −33 | −28 | −45 | −42 | −35 |
| Avg(ii) | −80 | −75 | −64 | −33 | −32 | −27 | −47 | −47 | −31 |

Note: Using two different metrics, this table shows the percentage change in risk exposures achieved by hedging a position in a particular sovereign bond using a position in one or more tranches of a 70:20:10 Sovereign Bond Backed Securitisation (three maturities are considered; 2-year, 5-year and 10-year). Issuers of the bonds are indicated at the beginning of each row as follows; AT (Austria), BE (Belgium), DE (Germany), ES (Spain), FI (Finland), FR (France), GR (Greece), IE (Ireland), IT (Italy), NL (Netherlands) and PT (Portugal). The first row for each case contains the percentage change in risk due to hedging where risk is measured as *standard deviation*. The second row for each case contains the percentage change in risk achieved through hedging where risk is measured as the range between the 5th and 95th quantiles of the returns distribution (implying VaR bounds). Columns of results permit a comparison of the risk reduction for three bond maturities and across three sub-sample periods.

*Appendix B.3. Hedge Effectiveness under Higher Incidence of Extreme Losses*

Table A3 compares hedge effectiveness using a t-copula rather than a Gaussian copula to trigger defaults. There is almost no difference for any sovereign (or on average) in the hedge effectiveness measures over the pre-crisis sample. The largest increase in the ratio of risks occurs for the case of GR (showing a very minor 4-point rise). For the Sovereign Debt Crisis and Recovery periods, a similar small change is apparent for all sovereign bonds except those of Germany (and Finland for the SDC period).

**Table A3.** Hedge effectiveness: Gaussian versus T-copula default trigger.

| | Pre-Crisis | | Sov Debt Crisis | | Recovery | |
|---|---|---|---|---|---|---|
| | **Gaussian** | **T-Dist** | **Gaussian** | **T-Dist** | **Gaussian** | **T-Dist** |
| AT(i) | −72 | −72 | −26 | −24 | −49 | −46 |
| AT(ii) | −84 | −83 | −41 | −38 | −56 | −52 |
| BE(i) | −77 | −77 | −20 | −19 | −52 | −49 |
| BE(ii) | −83 | −82 | −29 | −26 | −57 | −54 |
| DE(i) | −87 | −81 | −71 | −45 | −75 | −56 |
| DE(ii) | −89 | −83 | −73 | −48 | −74 | −53 |
| ES(i) | −69 | −68 | −28 | −34 | −43 | −40 |
| ES(ii) | −75 | −76 | −35 | −38 | −43 | −40 |
| FI(i) | −76 | −75 | −47 | −38 | −55 | −48 |
| FI(ii) | −84 | −83 | −55 | −44 | −62 | −55 |
| FR(i) | −83 | −84 | −31 | −26 | −59 | −57 |
| FR(ii) | −88 | −88 | −38 | −44 | −61 | −59 |
| GR(i) | −55 | −51 | −17 | −17 | −8 | 0 |
| GR(ii) | −67 | −63 | 23 | 29 | 12 | 27 |
| IE(i) | −52 | −50 | 1 | 0 | −27 | −26 |
| IE(ii) | −72 | −73 | −6 | −9 | −35 | −34 |
| IT(i) | −72 | −70 | −37 | −40 | −53 | −51 |
| IT(ii) | −77 | −71 | −44 | −40 | −54 | −55 |
| NL(i) | −81 | −81 | −43 | −37 | −56 | −51 |
| NL(ii) | −86 | −87 | −51 | −46 | −65 | −57 |
| PT(i) | −67 | −65 | 0 | 2 | −21 | −18 |
| PT(ii) | −77 | −76 | −9 | −14 | −26 | −21 |
| Avg(i) | −72 | −70 | −29 | −25 | −45 | −40 |
| Avg(ii) | −80 | −79 | −33 | −29 | −47 | −41 |

Note: Using two different metrics, this table shows the percentage change in risk exposures achieved by hedging a position in a particular sovereign bond using a position in one or more tranches of a 70:20:10 Sovereign Bond Backed Securitisation where the tranche yields have been estimated with a Gaussian and a t-Copula simulator. Issuers of the bonds are indicated at the beginning of each row as follows; AT (Austria), BE (Belgium), DE (Germany), ES (Spain), FI (Finland), FR (France), GR (Greece), IE (Ireland), IT (Italy), NL (Netherlands) and PT (Portugal). The first row for each case contains the percentage change in risk due to hedging where risk is measured as *standard deviation*. The second row for each case contains the percentage change in risk achieved through hedging where risk is measured as the range between the 5th and 95th quantiles of the returns distribution (implying VaR bounds). Columns of results permit a comparison of the risk reduction using two ways of estimating the tranche yields and across three sub-sample periods.

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
