# Peer review of "Positive Liquidity Spillovers from Sovereign Bond-Backed Securities"

_jrfm, doi:10.3390/jrfm12020058_

Reviewer 1 Report

This is a focused study on a narrow topic of interest to regulators, academics, and some market participants.

I find the analysis to be competent and without clear errors.  I suggest only minor revisions to make the topic of more generalizale interest.

Specific comments:

The last two sentences of the abstract are difficult to understand without first being familiar with the tests reported in the text.  I would suggest simplifying or more clearly explaining the what is being "assessed" about the hedge ratios and what is being modelled or implied about liquidity spillovers.  

I suggest that the author refrain from using the word "significantly" (last sentence of abstract) in the absence of a clearly specified statistical test.

The introduction could benefit from somewhat more background on SBBS proposals [3] and on the nature of the sovereign-bank doom-loop (line 28).

Table descriptions could be more clear.  In addition, it seems that simply reporting the difference, or reduction, in SDs could be a more intutitive metric to report in Tables 1-7, rather than the ratios of SDs.

It would be helpful to report the descriptive statistics on the raw data that are used as inputs for the simulations.  These distributions are essential to understanding the nature of the simulated output.

The figures need more descriptive captions.  In addition, many figures are redundant.

I enjoyed the paper and the approach but interest is likely to be very narrow.

Author Response

Many thanks to Reviewer 1 for these very helpful comments.  I outline below how each of these have been addressed and I hope that this is to the satisfaction of the reviewer.

Specific comment 1:

The last two sentences of the abstract are difficult to understand without first being familiar with the tests reported in the text.  I would suggest simplifying or more clearly explaining what is being "assessed" about the hedge ratios and what is being modelled or implied about liquidity spillovers.  I suggest that the author refrain from using the word "significantly" (last sentence of abstract) in the absence of a clearly specified statistical test.

Response:

Typically, hedges are only considered effective if they produce very large reductions in risk exposure.  In the paper this is shown by calculating the ratio of the hedged and unhedged variances (or Value-at-Risk bounds). These changes in risk exposure could indeed be assessed using statistical tests but these tests and I admit that not doing so looks a bit strange to most economics and finance researchers.  However, a priori, it is expected that the reduction in risk through hedging would be far greater than something marginal that needs to be statistically tested.  Hedgers typically would require reductions in risk that would be far in excess of reductions that marginally pass statistical tests.  The reviewers comment is nevertheless valid regarding how this is mentioned in the abstract.  I have therefore altered the abstract to the following…

This paper contributes to the debate concerning the benefits and disadvantages of introducing a European Sovereign Bond-Backed Securitisation (SBBS) to address the need for a common safe asset that would break destabilising bank-sovereign linkages. The analysis focuses on assessing the effectiveness of hedges incurred while making markets in individual euro area sovereign bonds by taking offsetting positions in one or more of the SBBS tranches. Tranche yields are estimated using the approach of [1]. Optimal hedging with SBBS is found to reduce risk exposures substantially in normal market conditions. In volatile conditions hedging is not very effective but leaves dealers exposed mostly to idiosyncratic risks. These remaining risks largely disappear if dealers are diversified in providing liquidity across country-specific secondary markets and SBBS tranches.

Specific comment 2:

The introduction could benefit from somewhat more background on SBBS proposals [3] and on the nature of the sovereign-bank doom-loop (line 28).

Response:

Two new paragraphs have been included to expand on this background as follows (just after the paragraph ending at line 30);

Traditionally, banks hold sovereign bonds as collateral for use in short term treasury management and in official monetary policy operations. This is encouraged by the fact that such assets are zero risk weighted for capital adequacy purposes. During the Great Recession and euro area Sovereign Debt Crisis many European banks experienced widespread defaults on their loan portfolios and they required recapitalization from their governments. Sovereigns simultaneously experienced severe current account deficits and expected imminent high bank bail-out costs which drove their yields to record levels. The higher risk of sovereign default fed back into credit and counter-party risks for the linked banks and this exacerbated the expected bail-out costs for sovereigns. These linked risk exposures generated what was referred to as a “vicious circle” at a euro area summit of heads of governments in 2012. It was also referred to as the “doom loop” by [5] or as the “diabolic loop” by [4].

Proposals to break the bank-sovereign vicious circle have been debated as part of a two-year reassessment by the Basel Committee on Banking Supervision of the Regulatory Treatment of Sovereign Exposures (RTSE) (see, [6]). The Committee decided to retain zero risk weights on domestic sovereign exposures acknowledging that there was no consensus among supervisors, experts, and economists on alternative policy options to amend the framework. Alternative approaches to breaking the doom loop have therefore appeared and the Sovereign Bond Backed Securitisation proposal by [4] is just one of these. Several related proposals are assessed by [7] and the analysis below would have some relevance for the ’Blue bond’ proposal discussed there since that proposal gives rise to a similarly liquid low risk bond that could be used in hedging by dealers.

Comment 3.

Table descriptions could be more clear.  In addition, it seems that simply reporting the difference, or reduction, in SDs could be a more intuitive metric to report in Tables 1-7, rather than the ratios of SDs.

Response:

The table descriptions have all been up-dated to fully reflect what is being presented in each case.  All ratios in the tables have been changed to percentage changes.  This necessitated some minor adjustments in the text as well where the results are discussed.

Comment 4.  

It would be helpful to report the descriptive statistics on the raw data that are used as inputs for the simulations. These distributions are essential to understanding the nature of the simulated output.

Response:

As regards the reasonableness of the sample period considered, the following paragraph was added on page 8... “The simulations that produce the SBBS yields are conducted on a daily basis over the period from soon after the introduction of the euro to the end of 2017. The simulations involve draws from 11 of the main sovereign bond yield-spreads (according to the report of the ESRB Task Force on Safe Assets, this group of sovereigns covers approximately 97.5% of outstanding sovereign debt in the euro area).  The time period covered contains a lot of variability in circumstances including; the pre-2008 Great Moderation period, the period of the financial crisis in the wake of the Lehman Brothers default, the euro area sovereign debt crisis of 2009-2012 and the subsequent gradual improvement in euro area sovereign bond markets - particularly those of peripheral member states. The later part of the sample also overlaps with implementation of unconventional monetary policy in the form of largescale bond purchases when these markets became less liquid and harder to hedge.”

To get a sense of the raw data used in the simulations a new figure was added showing the distribution of daily yield changes through time for the 11 sovereign bonds used in the simulations. In each case the GJR-GARCH(1,1) conditional volatility and the implied 1% VaR are displayed. Tabulating the descriptive stats was not so useful given the amount of variation in the yield data across a wide range of very different regimes. The following sentence is included with the above paragraph at the end of page 8…“Figure 1, provides a view of the data for the individual sovereign yield changes that was used in the SBBS yield simulation analysis. The negative of yield changes multiplied by 100 is shown. In each case the conditional volatility and 1% Value-at-Risk are also provided (where the volatility is estimated using a GJR-GARCH(1,1) and the latter is implied by the conditional volatility under normality). It is important to note that the scale of the y-axis is not constant across the panels of this figure.

Comment 5.

The figures need more descriptive captions.  In addition, many figures are redundant.

Response:

Captions for the figures have been extended to be more descriptive.

Comment 6.

I enjoyed the paper and the approach but interest is likely to be very narrow.

Response:

This could be the case but the debate about the introduction of a European safe asset is still very lively and this paper addresses one controversial argument that is very often made.  It is also worth noting that there has already been a citation of the working paper version of this article.

Reviewer 2 Report

The structure of the paper needs to be improved 1) There is need for a specific methodology section which explains why the sample is adequate and why the choice of the sample…. this is not clearly indicated. 2) I believe that in the Abstract line 6 the (1) needs to be opened up or explained in a short sentence.3) when referring to authors I believe you should mention the author and not just example as proposed by [2] -- it is quite annoying having to refer back and forth..
As for the rest it is well written and a good paper

Author Response

Many thanks to Reviewer 2 for these very helpful comments.  I outline below how each of these have been addressed and I hope that this is to the satisfaction of the reviewer.

Comment 1.

The structure of the paper needs to be improved. There is need for a specific methodology section which explains why the sample is adequate and why the choice of the sample…. this is not clearly indicated.

Response:

The structure has been improved by separating out parts that are clearly about methodology from parts that are strictly a discussion of results (I have also introduced some subsection divisions and changed the order of these slightly).  These occur on pages 8 to 11. 

I have included the following paragraph regarding the sample selection. “The simulations that produce the SBBS yields are conducted on a daily basis over the period from soon after the introduction of the euro to the end of 2017. The simulations involve draws from 11 of the main sovereign bond yield-spreads (according to the report of the ESRB Task Force on Safe Assets, this group of sovereigns covers approximately 97.5% of outstanding sovereign debt in the euro area).  The time period covered contains a lot of variability in circumstances including; the pre-2008 Great Moderation period, the period of the financial crisis in the wake of the Lehman Brothers default, the euro area sovereign debt crisis of 2009-2012 and the subsequent gradual improvement in euro area sovereign bond markets - particularly those of peripheral member states. The later part of the sample also overlaps with implementation of unconventional monetary policy in the form of largescale bond purchases when these markets became less liquid and harder to hedge.”

Comment 2.

I believe that in the Abstract line 6 the (1) needs to be opened up or explained in a short sentence.

Response:

This is a fair comment and to address this the following sentence has been added to the abstract. “This involves simulation of defaults and allocation of the combined credit risk premium of the components to the SBBS tranches according to the seniority of claims.”

Comment 3.

When referring to authors I believe you should mention the author and not just example as proposed by [2] -- it is quite annoying having to refer back and forth.. 

Response:

I agree with this and will ask MPDI if it would be possible to get the cites to work this way (I was unable to get this to work using the MPDI latex format).

Comment 4.

As for the rest it is well written and a good paper.

Response:

None required.